# Isolation Strategy towards Earth-Abundant Single-Site Co-Catalysts for Photocatalytic Hydrogen Evolution Reaction

**Pablo Ayala** [1] , **Ariane Giesriegl** [1] , **Sreejith P. Nandan** [1] , **Stephen Nagaraju Myakala** [1], **Peter Wobrauschek** [2] **and Alexey Cherevan** [1,*]

[1] Institute for Materials Chemistry, TU Wien, 1060 Vienna, Austria; pablo.ayala@tuwien.ac.at (P.A.); ariane.giesriegl@tuwien.ac.at (A.G.); sreejith.nandan@tuwien.ac.at (S.P.N.); stephen.myakala@tuwien.ac.at (S.N.M.)

[2] Atominstitute, TU Wien, 1020 Vienna, Austria; peter.wobrauschek@tuwien.ac.at

* Correspondence: alexey.cherevan@tuwien.ac.at; Tel.: +43-58801-165230

**Abstract:** Achieving efficient photocatalytic water splitting remains one of the most vital challenges in the photocatalysis field, as the performance of contemporary heterogeneous catalysts is still limited by their insufficient activity and low predictability. To address this challenge, this work takes inspiration from the concept of heterogeneous single-metal-site catalysts (HSMSCs) and follows the idea of site-isolation, aiming towards single-site co-catalyst species and a higher atom-utilization efficiency. We synthesized a set of photocatalysts through an adsorption-limited wet impregnation process using bare and phosphate-modified $TiO_2$ as model supports and earth-abundant metals (Cu and Ni) with various loadings (0.008–5 wt.%) as co-catalyst species. The catalysts are characterized by TXRF for the determination of the real co-catalyst loadings, UV-vis and FTIR spectroscopes for semi-quantitative analysis of the metal state and binding modes to the substrate, and HRTEM for resolving the morphology of the sample's surface. All samples were then evaluated towards the photocatalytic hydrogen evolution reaction (HER). We show that much higher turnover frequencies (TOFs) are obtained for both Cu- and Ni-based systems when lower co-catalyst loadings are used, which indicates an improved atom-utilization efficiency that reaches performances comparable to the noble Au co-catalyst. We also introduce a structural model to explain the observed TOF trends, which confirms that both earth-abundant systems undergo a strong structural reconstruction upon site-isolation towards smaller, perhaps even single-site-like species.

**Keywords:** single-site; single-atom; water splitting; heterogeneous single-metal-site catalysts; single-atom-catalysis (SAC); photocatalysis; site-isolation; copper; nickel; co-catalyst; HER



## 1. Introduction

Photocatalysis allows the transformation of solar to chemical energy by facilitating non-spontaneous chemical reactions with the aid of light and a photocatalyst. Through photocatalysis, the photon's energy can be stored in the chemical bonds of solar fuels—commodity chemicals that can be generated from nothing but sunlight and abundant feedstocks through photocatalytic water splitting and/or carbon dioxide photoreduction [1,2]. Both reactions, however, involve kinetically complex redox processes which require a delicate design of the catalytic sites. In this light, the use of structurally well-defined and thus more selective co-catalysts has shown much promise to address this challenge [3–5]. The construction of advanced co-catalysts remains not-trivial, and new synthetic and methodological approaches still need to be established to allow for purposeful design of effective and selective photo- and co-catalysts towards energy conversion and storage.

In this respect, heterogeneous single-metal-site catalysts (HSMSCs) offer many advantages over their bulkier counterparts as a result of their ability to bridge the qualities of homo- and heterogeneous catalysis [6,7]. When relying on the concept of HSMSCs, maximized atom utilization efficiency and accessibility of the active sites (homogeneous traits)

render excellent catalytic performances, while stability and recyclability (heterogeneous traits) remain preserved. Note that the exact definition of "single-site" has been debated in the literature, thus to avoid confusions, and this work will follow a broad description given by Zecchina et al. for the single-site to be "a metal atom, ion or small cluster of atoms held by surface ligands to a rigid framework" [8].

HSMSCs and single-atom-catalysts (SACs) have both been widely explored in conventional heterogeneous catalysis, however, applying these concepts in photocatalysis is a rather novel approach and only a few excellent systems have so far been reported [9]. We note, however, that the approach has so far been mostly applied to noble-metal systems [10–16] and thus remains underexplored towards earth-abundant co-catalysts (so far mainly Co and Cu [17–23]), which can enable the synthesis of more unique and selective co-catalysts able to address the challenges of the abovementioned complex multi-electron catalytic reactions [24].

A number of synthetic strategies towards HSMSC have been reported [25], from which the importance of support surface chemistry (e.g., defects and coordination-sites) have been highlighted to benefit selective and irreversible single-site anchoring. Besides, the spatial isolation of metal active sites has also been shown to inhibit the undesired growth of the single-site catalyst species [26], which may further benefit the stability of co-catalysts under photocatalytic conditions.

In this work, to complement contemporary efforts towards single-site photocatalysis, we explore a combination of HSMSC, surface-tailoring, and site-isolation strategies aiming to construct a set of earth-abundant photocatalysts based on Ni and Cu and promote the formation of single-site co-catalyst species. HER performance trends suggest that strongly enhanced turnover frequencies achieved at lower co-catalyst loadings, which indicates a much more efficient co-catalyst-site utilization and ultimately highlights the success of the isolation strategy and surface-modification.

## 2. Results and Discussion

### 2.1. The Site-Isolation Strategy

The photocatalyst preparation method followed the site-isolation strategy shown in Figure 1, which assumes that the formation of single-sites is more likely to be achieved by isolating the precursor species on the support's surface and by suppressing their movement (e.g., via surface diffusion) and potential growth. To probe the effect of site-isolation experimentally, we varied the initial co-catalyst precursor concentration within 3 orders of magnitude, ranging from 5 to 0.008 wt.% with respect to $TiO_2$ support and followed an adsorption-limited wet impregnation process to load $Ni^{2+}$ and $Cu^{2+}$ species onto the surface of model anatase nanoparticles (synthetic details can be found in Methods). Higher co-catalyst loadings (Figure 1b) allow the neighboring sites to interact, either during the adsorption process or in the consequent catalytic reaction, which could potentially lead to the formation of nanoparticles. However, when going towards sufficiently low loadings and thus surface coverage (Figure 1c), only a small number of co-catalyst precursor atoms is deposited on each $TiO_2$ particle, which are stabilized by the most suitable surface defects of the support. As a result, undesired co-catalyst growth can be hindered, and only small clusters, or ideally single-co-catalyst-sites, can be achieved [25]. To further examine the importance of substrate surface chemistry, we compared bare $TiO_2$ with its $PO_4$-modified counterpart (denoted as $PO_4/TiO_2$, synthesis in Methods). Two series of photocatalysts were prepared: $x/Ni/TiO_2$ and $x/Cu/TiO_2$ (x representing the intended metal loading in wt.%), as well as $x/Ni/PO_4/TiO_2$ and $x/Cu/PO_4/TiO_2$.

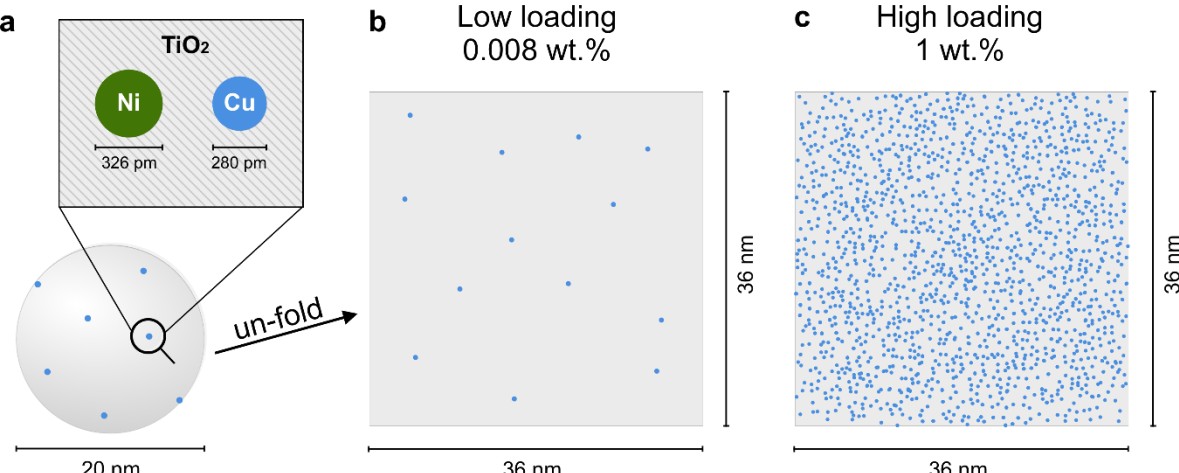

**Figure 1.** Graphical representation of the site-isolation strategy showing: (**a**) single TiO$_2$ nanoparticle and scaled Cu and Ni atoms (the representation is based on the surface area values), as well as its unfolded surface for (**b**) low co-catalyst loading of 0.008 wt.% (13 Cu atoms) and (**c**) high co-catalyst loading of 1 wt.% (1586 Cu atoms).

## 2.2. Characterization

### 2.2.1. Real Co-Catalyst Loadings

Our synthetic protocol involves several washing steps after the initial TiO$_2$ impregnation to only allow strongly adsorbed, e.g., defect-stabilized, co-catalyst species to stay on the support's surface (synthetic details in Methods). Thus, to characterize the real co-catalyst loadings in the resulting composites, we performed elemental analyses of the as-synthesized photocatalyst powders using highly sensitive total reflection X-ray fluorescence spectroscopy (TXRF). Figure 2a shows the intended vs. the real loadings (note the logarithmic scale) of Cu and Ni on TiO$_2$ and PO$_4$/TiO$_2$. The red dashed line represents the maximum achievable loading values, i.e., intended loadings. A deviation from this line means that precursor species were removed from the support's surface during the washing process, resulting in lower real loadings (Figure S1). The summary data in Table 1 suggest that only 5–20% of the precursor species adsorb on the bare TiO$_2$ surface strongly enough, while the introduction of phosphate groups increases this value significantly, especially for higher intended loading values (black arrows in Figure 2a). The strong deviation of the real loadings from the intended values is likely related to the high solubility of Cu$^{2+}$ and Ni$^{2+}$ in water and their tendency to undergo ligand exchange and form aqua complexes, which may facilitate their detachment from the surface. As our experiments demonstrate (Figure S2), the nature of the precursor (i.e., the choice of ligands) does affect this surface-solution equilibrium and thus makes a pronounced effect on the metal loading values. The effect of the TiO$_2$ surface modification towards higher loadings for the PO$_4$/TiO$_2$ support, on the other hand, provides a strong confirmation that an increase in surface charge and the abundance of complementary adsorption sites are important factors to stabilize the co-catalyst species.

### 2.2.2. Adsorption and State of Cu and Ni

Attenuated total reflection Fourier-transform infrared spectroscopy (ATR-FTIR) spectra of both catalyst series were first taken to elucidate the Cu/Ni binding modes after impregnation (Figure S3). For bare TiO$_2$ support, the broad band centered at around 3350 cm$^{-1}$ is characteristic for vibrational modes of the hydroxyl groups (-OH) of weakly adsorbed water molecules at the surface of TiO$_2$, while the sharp peak at 1634 cm$^{-1}$ can be attributed to the -OH stretching mode of the surface hydroxyls, caused by the dangling bonds of TiO$_2$. Upon impregnation with Cu and Ni precursor, both OH-specific modes (Figure 2b,c) decrease in intensity, which indicates that Cu$^{2+}$ and Ni$^{2+}$ species (acting as Lewis acids) bind strongly to the O atoms (acting as Lewis base) of the TiO$_2$ surface hydroxyls.

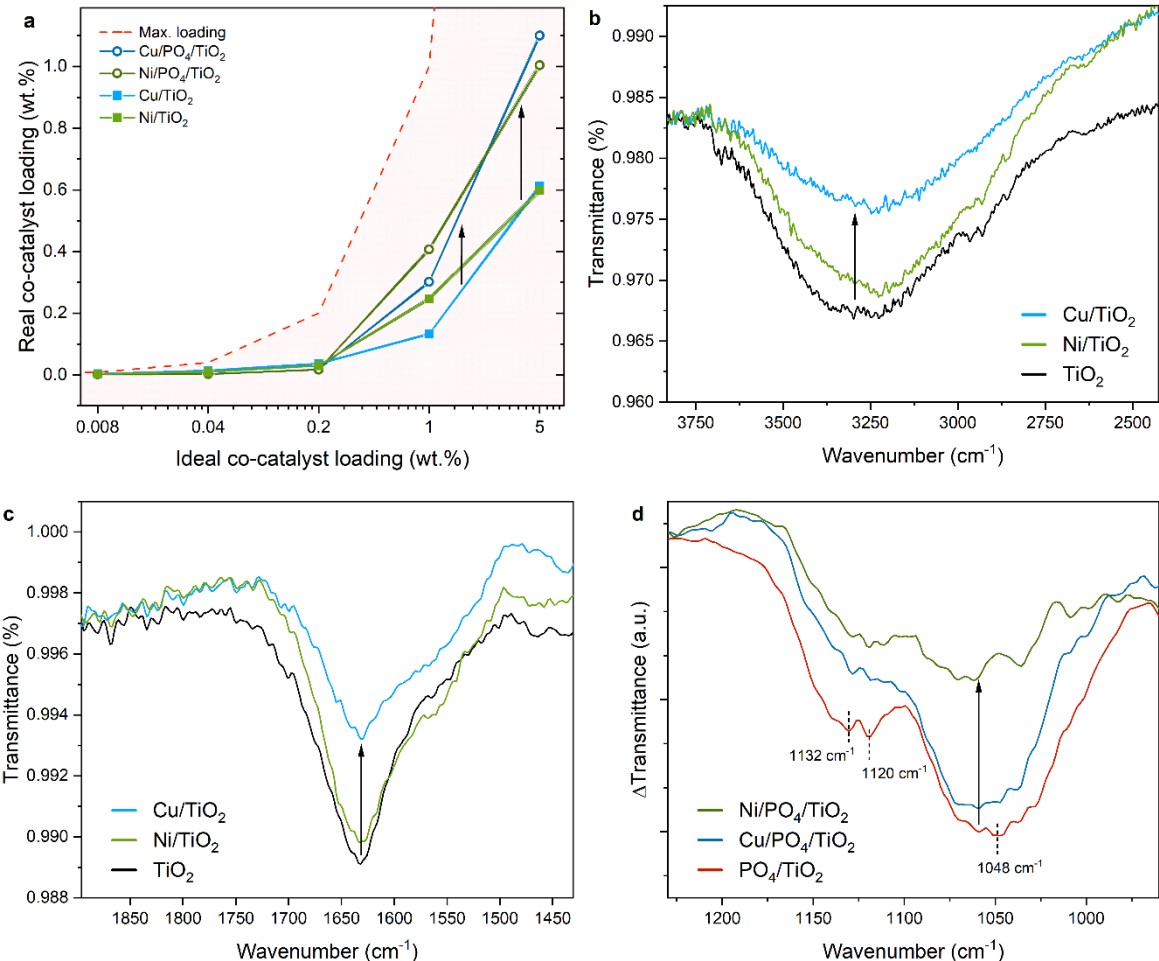

**Figure 2.** (**a**) Comparison of intended vs. real loadings of the different co-catalysts for Cu and Ni on bare $TiO_2$ and $PO_4/TiO_2$ based on TXRF data and ATR-FTIR spectra: (**b**) OH-specific broad band at 3350 cm$^{-1}$ for bare $TiO_2$ (black) 5/Cu/$TiO_2$ (blue) and 5/Ni/$TiO_2$ (green), (**c**) OH-specific mode at 1634 cm$^{-1}$ for bare $TiO_2$ (black) 5/Cu/$TiO_2$ (blue) and 5/Ni/$TiO_2$ (green), and (**d**) $PO_4$-specific modes for bare $PO_4/TiO_2$ (red), 5/Cu/$PO_4/TiO_2$ (blue) and 5/Ni/$PO_4/TiO_2$ (green), the spectra in d are obtained by subtracting FTIR data of $TiO_2$ (Δ Transmittance in a.u. = arbitrary units).

In the case of the $PO_4/TiO_2$ support, ATR-FTIR, first of all, provides a confirmation of the successful surface modification with phosphate, which is evident by a pronounced $PO_4$-related doublet in the region between 1175 and 970 cm$^{-1}$ (Figure S4) [27–29]. For the $Cu/PO_4/TiO_2$ and $Ni/PO_4/TiO_2$ samples, the difference in the $TiO_2$-related OH-specific modes after impregnation is not as prominent as for the samples prepared on bare $TiO_2$ (Figure S3b). However, Figure 2d indicates the decrease of the $PO_4$-related band upon impregnation, which suggests that adsorption of Cu and Ni proceeds almost exclusively via surface $PO_4$ groups.

Upon $Cu^{2+}$ and $Ni^{2+}$ precursor adsorption, the cations are likely to be ligated by the hydroxyl-groups of the $TiO_2$ surface, staying in their ionic form. However, $TiO_2$ is also known to promote photoreduction of metals in the solution [30], which may lead to a more complex situation given the reasonably low affinity of both Cu and Ni to O [31]. Thus, to elucidate the nature of the as-impregnated co-catalyst species, we next acquired absorption spectra of the photocatalyst powders through diffuse-reflectance spectroscopy (DRS). In the case of $Cu/TiO_2$ composites (Figure 3a and Figure S5) the absorption profiles exhibit a pronounced broad peak centered at around 800 nm, which corresponds well to a d-d transition of molecular $Cu^{2+}$ complexes at higher wavelengths [32]. The intensity of this peak increases with increasing Cu loadings, suggesting that Cu is present in the system in its $Cu^{2+}$ form at all loading values. An exception is found for the highest Cu loading,

at which a pronounced shoulder at 425 nm appears. This could be ascribed to the surface plasmon resonance (SPR) bands of Cu nanoparticles and can be related to partial sintering of the excess Cu species due to their high content and proximity (Figure 1b), possibly via photoreduction after initial surface-adsorption takes place [33].

**Table 1.** Summary data of the achieved real loadings, determined through TXRF for Cu (above) and Ni (below) and both $TiO_2$ and $PO_4/TiO_2$ support series.

| $Cu/TiO_2$ | Real Loading (wt.%) * | $Cu/PO_4/TiO_2$ | Real Loading (wt.%) |
|---|---|---|---|
| $5/Cu/TiO_2$ | $0.6126 \pm 0.0072$ | $5/Cu/PO_4/TiO_2$ | $1.1004 \pm 0.0090$ |
| $1/Cu/TiO_2$ | $0.1326 \pm 0.0036$ | $1/Cu/PO_4/TiO_2$ | $0.3012 \pm 0.0048$ |
| $0.2/Cu/TiO_2$ | $0.0366 \pm 0.0024$ | $0.2/Cu/PO_4/TiO_2$ | $0.0180 \pm 0.0018$ |
| $0.04/Cu/TiO_2$ | $0.0144 \pm 0.0018$ | $0.04/Cu/PO_4/TiO_2$ | $0.003 \pm 0.0012$ |
| $0.008/Cu/TiO_2$ | $0.0036 \pm 0.0018$ | $0.008/Cu/PO_4/TiO_2$ | <0.0018 ** |
| $Ni/TiO_2$ | Real Loading (wt.%) * | $Ni/PO_4/TiO_2$ | Real Loading (wt.%) |
| $5/Ni/TiO_2$ | $0.5976 \pm 0.0084$ | $5/Ni/PO_4/TiO_2$ | $1.0038 \pm 0.0102$ |
| $1/Ni/TiO_2$ | $0.2472 \pm 0.0054$ | $1/Ni/PO_4/TiO_2$ | $0.4068 \pm 0.0060$ |
| $0.2/Ni/TiO_2$ | $0.0429 \pm 0.0043$ | $0.2/Ni/PO_4/TiO_2$ | $0.0168 \pm 0.0018$ |
| $0.04/Ni/TiO_2$ | $0.0096 \pm 0.0018$ | $0.04/Ni/PO_4/TiO_2$ | $0.0024 \pm 0.0002$ |
| $0.008/Ni/TiO_2$ | $0.0024 \pm 0.0018$ | $0.008/Ni/PO_4/TiO_2$ | <0.0018 ** |

* Standard deviation values for each measurement are indicated with $\pm$. ** Maximal values are indicated due to reaching the detection limits of the spectrometer.

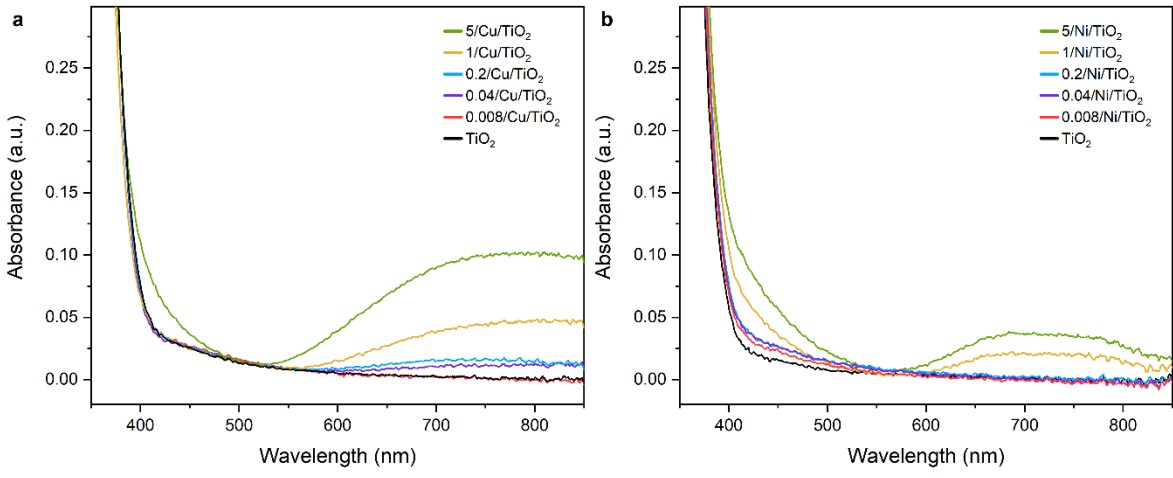

**Figure 3.** Absorbance spectra obtained through DRS measurements of (**a**) $Cu/TiO_2$ and (**b**) $Ni/TiO_2$ composite series.

The absorption spectra of the $Ni/TiO_2$ series suggest a more complex scenario. Figure 3b shows the appearance of a shoulder at 410–520 nm, which gets more pronounced for higher loadings (1 and 5 wt.%). This shoulder can be assigned to previously reported Ni-doping [34] or to the formation of Ni nanoparticles [35]. The broad peak between 590 and 860 nm corresponds well to $Ni^{2+}$ species coordinated by the acetylacetonate ligands [36] or, more generally, to $Ni^{2+}$ complex species coordinated by aqua or hydroxo ligands that form after adsorption and subsequent ligand-exchange of the Ni precursor [37,38]. However, metallic Ni nanoparticles have also been reported to feature a broad absorption starting at 700 nm [39]. Hence, we assume that the generated $Ni/TiO_2$ catalysts—at least for higher concentrations—have a mixed nature, in which Ni-based nanoparticles and ionic $Ni^{2+}$ species coexist. This is possibly driven by the heterogeneity of the $TiO_2$ surface structures and a partial growth of $Ni^0$ species, caused by the photo-reductive properties of $TiO_2$.

To complement these data, we further performed X-ray photoelectron spectroscopy (XPS) studies, however, due to the low co-catalyst content in both photocatalyst series, even at the highest real loadings (1.0 wt.% for $Ni/PO_4/TiO_2$ and 1.1 wt.% for $Cu/PO_4/TiO_2$), the interpretation of the obtained Cu and Ni signals does not allow for any reliable conclusion regarding the oxidation state of the metals.

### 2.2.3. Morphology and Structure

Next, we evaluated the structural and morphological properties of the prepared photocatalyst series. Scanning electron microscopy (SEM) images (Figure S6) acquired for both composite series indicated no change in the sample's morphology upon co-catalyst impregnation. High-resolution transmission electron microscopy (HRTEM) data for the exemplarily $5/Cu/TiO_2$ sample (Figure 4a) gives no indication for surface-decorating co-catalyst nanoparticles on the $TiO_2$ surface. This result is in line with DRS, thus further confirming that Cu stays in its $Cu^{2+}$ form after impregnation and builds surface-attached layers or clusters, not distinguishable with this instrument's resolution. In contrast, TEM images for the $5/Ni/TiO_2$ sample in Figure 4b show small foreign nanoparticles between 2.2 and 2.5 nm decorating the $TiO_2$ surface. This observation corresponds to the conclusion of the DRS data, from which the presence of Ni nanoparticles was suggested for higher loadings. TEM investigations of the samples with lower co-catalyst loadings (<1 wt.%) did not show any formation of Ni- or Cu-based nanoparticles (Figure S7). This further strengthens the success of the site-isolation strategy, which is supposed to prevent sintering and growth of the surface-immobilized species for lower loading values. Complementarily, TEM images of the $5/Cu/PO_4/TiO_2$ and $5/Ni/PO_4/TiO_2$ samples (Figure 4c,d) do not show any metal nanoparticles either, although TXRF data demonstrated relatively higher loadings after the substrate modification. This observation strongly suggests that the phosphate layer does increase the strength of the adsorption interactions between the support and co-catalyst, which consequently limits surface diffusion of the metal species and prevents the formation of large aggregates.

Powder X-ray diffraction (XRD) with a Si internal standard was performed to gain information regarding present phases, lattice parameters, and crystallinity. The spectra of the as-prepared samples, shown in Figure 5a, only feature peaks characteristic of anatase $TiO_2$, and no additional signals related to Ni- or Cu-based compounds (e.g., oxide nanoparticles or metal-doped $TiO_2$ phases) could be found. Detailed refinement of the $TiO_2$ reflections further suggests that even for the samples with the highest co-catalyst loadings, e.g., $5/Ni/TiO_2$ and $5/Cu/TiO_2$, the anatase crystallinity characteristics (such as lattice parameters in Table 2 [40,41]) stay preserved and no significant variation between different samples is introduced. The same situation is observed for the $PO_4/TiO_2$ series (Figure 5b). These results support the idea of surface immobilization of the co-catalyst species and speak against significant Cu or Ni-doping into the $TiO_2$ lattice.

### 2.3. Photocatalytic Performance

All photocatalytic systems were investigated towards the photocatalytic hydrogen evolution reaction in water–methanol mixtures following protocols widely accepted in the community (experimental details can be found in Methods). A summary shown in Figure 6a exemplarily demonstrates the absolute rates of $H_2$ evolution for the Cu and Ni series on bare $TiO_2$ plotted against experiment time. Analogous experiments were performed for the $PO_4/TiO_2$ series. For all $Cu/TiO_2$ samples, the rates reach a plateau, which corresponds to a stable photocatalytic performance. For all $Ni/TiO_2$ samples, we observed a gradual HER activation, which is in line with the in situ reduction of $Ni^{2+}$ species reported recently [24]. Besides this, the catalytic performance of Cu co-catalysts is generally greater than that of Ni, which goes in line with other studies [42–44] and is likely related to the more favorable HER kinetics of the former [45].

Importantly for the site-isolation strategy, we observed an expected decrease in absolute HER rate values with decreased intended loadings of Cu or Ni on the respective

support. However, these absolute values cannot be compared with each other due to the lack of a normalization per amount of co-catalyst site. Since the main motivation of this work is to evaluate the impact of the site-isolation strategy on the atom-utilization efficiency, the following section will discuss these data in the context of turnover frequency (TOF) values (see Methods for details).

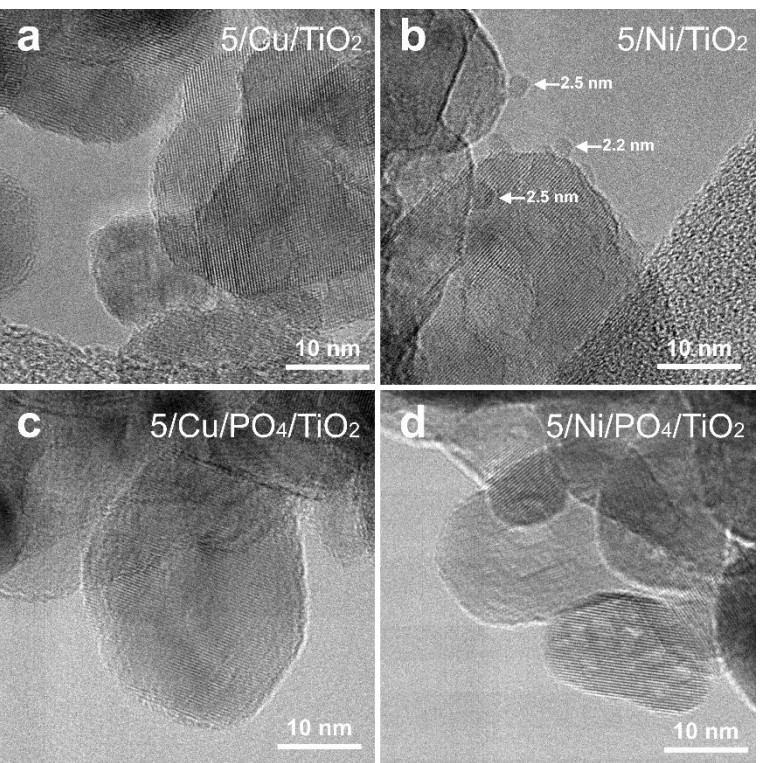

**Figure 4.** HRTEM images for the composites: (**a**) $5/Cu/TiO_2$, (**b**) $5/Ni/TiO_2$, (**c**) $5/Cu/PO_4/TiO_2$ and (**d**) $5/Ni/PO_4/TiO_2$.

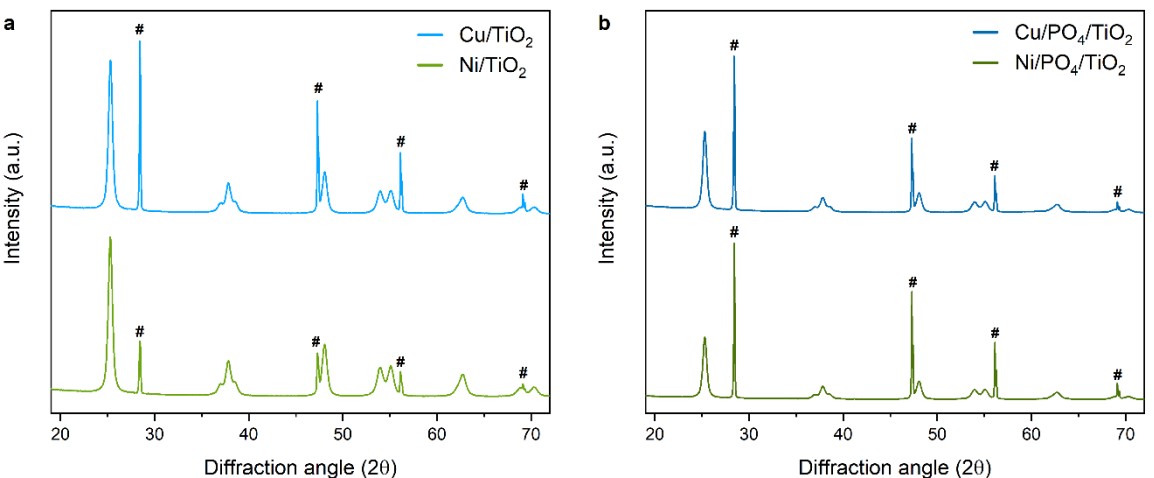

**Figure 5.** PXRD spectra of (**a**) $5/Cu/TiO_2$ and $5/Ni/TiO_2$ as well as (**b**) $5/Cu/PO_4/TiO_2$ and $5/Ni/PO_4/TiO_2$ samples. Peaks indicated with # correspond to Si.

**Table 2.** Lattice parameters of the $TiO_2$ anatase phase in different samples.

| Sample | Si * (a, Å) | Anatase (a, Å) | Anatase (c, Å) |
|---|---|---|---|
| $5/Cu/TiO_2$ | 5.431 | 3.789 | 9.495 |
| $5/Ni/TiO_2$ | 5.431 | 3.789 | 9.511 |
| $5/Cu/PO_4/TiO_2$ | 5.431 | 3.788 | 9.511 |
| $5/Ni/PO_4/TiO_2$ | 5.431 | 3.789 | 9.511 |

* Silicon was used as an internal standard.

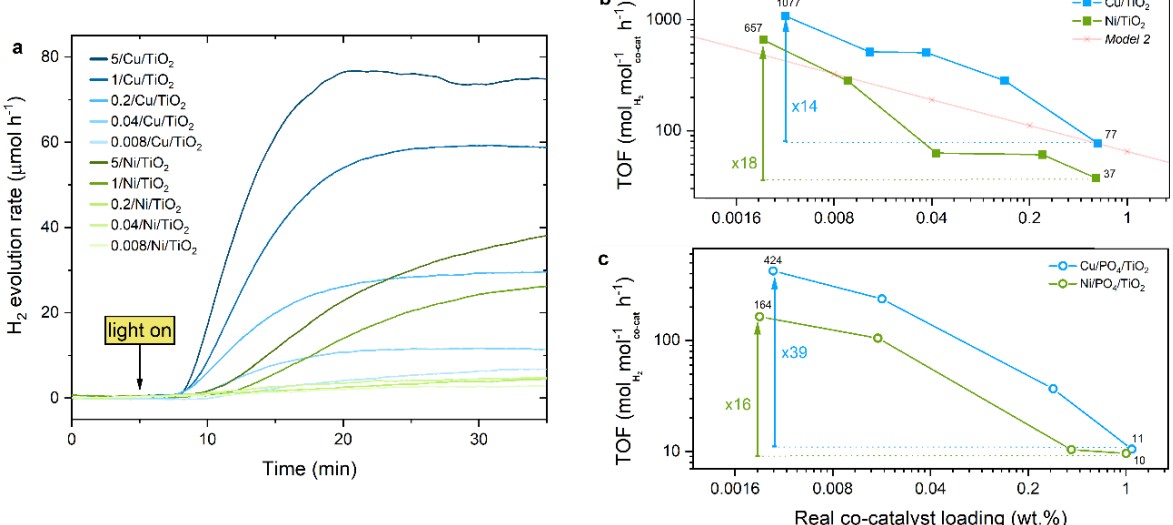

**Figure 6.** Photocatalytic performance: (**a**) Exemplary HER profiles for $Cu/TiO_2$ and $Ni/TiO_2$ series and (**b**) real co-catalyst loadings against maximum HER-activity per mol co-catalyst (expressed in TOF values) for $TiO_2$ and (**c**) $PO_4/TiO_2$ series. Red dashed line represents the Model 2 fit (see Appendix A).

### 2.3.1. TOF Evaluation

The turnover frequency allows a better comparison of all individual samples in the respective series because it takes the actual amount of co-catalyst responsible for the catalytic reaction into account. By plotting the real loadings (derived from TXRF data) of the Cu- and Ni-based photocatalysts against the maximum HER-activity values normalized per amount of catalytically active species, Figure 6b,c are obtained for the $TiO_2$ and $PO_4/TiO_2$ series, respectively. These plots demonstrate that the TOF values, i.e., moles of produced $H_2$ per moles of co-catalyst per hour, grow strongly when lower co-catalyst loadings are used. Considering $Cu/TiO_2$ as an example, the $5/Cu/TiO_2$ composite (real loading corresponds to 0.613 wt.%, see Table 1), a TOF of 77 $h^{-1}$ can be obtained, which means that 77 molecules of $H_2$ can be produced by each of the Cu atoms of this photocatalyst per hour. Going into lower intended (and thus real) loadings, the TOF numbers increase to 283 (for $1/Cu/TiO_2$) and 505 (for $0.2/Cu/TiO_2$) and 512 (for $0.04/Cu/TiO_2$). Finally, the highest TOF of 1077 $h^{-1}$ is obtained for the $0.008/Cu/TiO_2$ sample (real loading corresponds to 0.0036). Overall, we see an improvement factor of 14 between the highest and lowest Cu loadings, which manifests a strongly enhanced atom utilization efficiency. A similar trend is observed for $Ni/TiO_2$ series with a maximal TOF value of 657 $h^{-1}$ and an overall improvement factor of 18.

Importantly, these TOF values are in a similar order of magnitude as the highest value for the reference $Au/TiO_2$ (2576 $h^{-1}$) prepared following a similar method (Figure S8). Considering the scarcity of noble metals, our isolation strategy protocol of earth-abundant co-catalysts becomes especially attractive.

### 2.3.2. Geometrical Models

In order to gain more insights into the effect of the site-isolation strategy followed in this work, we developed two geometrical models (detailed description in Appendix A). Model 1 describes the behavior in which the shape/size of the co-catalyst does not change upon decreasing the amount of precursor deposited. A lower catalyst deposition merely results in a proportionally lower number of catalytically active sites (be it surface atoms of clusters or small nanoparticles). Based on this model, we would expect a constant trend for loading vs. TOF upon loading decrease, shown as a purple line in Figure A2. In contrast, Model 2 describes the scenario in which the co-catalyst shape/size does change upon a decrease of the co-catalyst loading. Assuming the original co-catalyst nanoparticles (likely to be formed at the highest loadings) are perfect spheres, in which the number stays unchanged but sizes are reduced upon loading decreases, we would expect the loading vs. TOF to follow the negative exponential function shown as a red line in Figure A2.

Comparison of the $Cu/TiO_2$ and $Ni/TiO_2$ profiles with the models allows us to make several qualitative conclusions. First, as the experimental TOFs increase strongly for lower co-catalyst loadings and a non-linear trend of this increase is observed (similar to Model 2, see dashed red line in Figure 6b), the data suggest that an evolution of the shape/size of the co-catalyst species takes place indeed. Second, as the values of the experimentally measured TOF improvement factors surpass those predicted by Model 2, the data for $Cu/TiO_2$ and $Ni/TiO_2$ series further suggest an even stronger morphological change of the co-catalyst species and a stabilization of eventually smaller clusters and single-sites.

### 2.3.3. Effect of the Support

Figure 6c sheds further light on the impact of substrate chemistry on the success of the site-isolation strategy. The relative increase of expected TOF values between the highest and the lowest loadings is partially higher for the $PO_4/TiO_2$ support compared to that of the bare $TiO_2$, with a factor of 39 and 16 reached for $Cu/PO_4/TiO_2$ and $Ni/PO_4/TiO_2$, respectively. This is in line with the improved adsorption capacity of the $PO_4$-modified $TiO_2$ but—as TOF values are normalized per co-catalyst mass—it also suggests that stronger-binding phosphate groups provide an additional stabilization effect. It restricts surface diffusion of Cu and Ni species and limits their growth, thus ultimately resulting in enhanced stabilization of smaller single-co-catalyst-site species.

Moreover, the Cu and Ni on $PO_4/TiO_2$ photocatalysts become comparable to the reference $Au/PO_4/TiO_2$ series (Figure S8). At the lowest loadings, Cu even surpasses Au, reaching a maximal TOF value of $424\ h^{-1}$ compared to the $383\ h^{-1}$ of its noble counterpart.

## 3. Materials and Methods

### 3.1. List of Chemicals

All materials used for the syntheses were obtained from commercial suppliers. Table 3 contains information on the identification, purity and brand of the used chemicals. All of them were used without further purification.

### 3.2. Synthesis of Phosphate-Modified TiO$_2$

The phosphate modification of $TiO_2$ was performed through a wet impregnation process, similar to the ones reported by Davydov, Zhao and Liu [27–29]. $TiO_2$ (2 g, 25 mmol) was dispersed in 300 mL of DI water by sonicating for 10 min at room temperature. 200 mL of a 12.2 mM phosphoric acid solution was added to the dispersion under stirring, aiming a monolayer of $PO_4$ groups attaching to the support's surface. To assure the full coverage, we used an excess amount of $PO_4^{3-}$ ions (2.45 mmol), which take up an area 2.13 times the surface area of the support used for this batch. This factor was calculated considering the BET surface area ($78.6\ m^2\ g^{-1}$) of $TiO_2$ and the apparent ionic radius of $PO_4^{3-}$ of 238 pm [46,47]. The dispersion was stirred for 1 h at 500 rpm, then sonicated again for 10 min to homogenize the dispersion components and facilitate phosphate ions reaching the particles' surface. The dispersion was stirred for a further 4 h at room temperature

to ensure complete adsorption of phosphate ions onto the surface. Subsequently, the phosphate-modified $TiO_2$ was filtered and washed thoroughly with DI water until the filtrate was of a neutral pH. The washing solutions were saved for further analysis. For drying, the wet filter cake was put into the oven overnight at 60 °C. Finally, the dry sample was mortared and given the name $PO_4/TiO_2$.

**Table 3.** Information on the used chemicals.

| Chemical Formula | Name | CAS Number | Purity | Brand |
|---|---|---|---|---|
| $CH_3OH$ | Methanol | 67-56-1 | absolute | VWR chemicals |
| $TiO_2$ | Anatase | 1317-70-0 | 99.7% | Sigma-Aldrich |
| $H_3PO_4$ | Phosphoric acid | 7664-38-2 | extra pure | Acros organics |
| $Ni(Ac)_2 \cdot 4H_2O$ | Nickel(II) acetate tetrahydrate | 6018-89-9 | 99% | Sigma-Aldrich |
| $Ni(AcAc)_2$ | Nickel(II) acetylacetonate | 3264-82-2 | 96% | Sigma-Aldrich |
| $Cu(Ac)_2 \cdot H_2O$ | Copper(II) acetate monohydrate | 6046-93-1 | 99% | Fluka |
| $HAuCl_4 \cdot 3H_2O$ | Tetrachloroauric(III) acid trihydrate | 16961-25-4 | 99.99% | Alfa Aesar |

*3.3. Co-Catalyst Modification of TiO₂ (Impregnation Process)*

The precursors selected for the transition metal modification were metal acetate (Ac) hydrates ($Ni(Ac)_2 \cdot 4H_2O$ and $Cu(Ac)_2 \cdot H_2O$) and for nickel, an additional acetylacetonate (AcAc) precursor was used: $Ni(AcAc)_2$. The precursor selected for the noble metal reference systems was $HAuCl_4$. Using the metal acetates and nitrates, stock precursor solutions ($c = 10$ g $L^{-1}$) were prepared in DI water. Due to solubility problems, the stock solution for $Ni(AcAc)_2$ was 0.2 g $L^{-1}$. The stock solutions for the $HAuCl_4$ (1 g $L^{-1}$ in water) were prepared under argon and kept under dark to prevent undesired nanoparticle formation.

For the impregnation process, 50 mg of the selected substrate material ($TiO_2$ or $PO_4/TiO_2$) was suspended in 50 mL of DI water by sonicating for 10 min to break down the eventually agglomerated nanoparticles and expose their surface area for easier accessibility of the precursors. The desired volume of the precursor solution was consequently added to the substrate suspension, calculated depending on which ideal loading was aimed for. The investigated metal loadings were between 0.008 and 5 wt.%. To secure a proper dispersion of the precursor, the suspension was sonicated again for a further 5 min. The suspension with the precursor was then stirred for 1 h at 550 rpm to secure complete adsorption at the surface of the support. The impregnated product was then filtered and washed 3 times with 50 mL of DI water to remove weakly bound precursor species (both cations and anions). The washing solutions were saved for further elemental analysis. Finally, the wet filter cakes were dried overnight in the oven at 60 °C. A set of control samples of the bare supports were also prepared following the same procedure but not adding any precursor solution. The general naming of these "composite materials" works as follows: wt.%/metal/support. As an example, adding 5 wt.% of copper to the phosphate modified $TiO_2$ would result in a sample referred to as $5/Cu/PO_4/TiO_2$. It must be noted that real loadings vary from ideal ones since precursor metal species are being washed away from the surface. However, the naming shall account for ideal (intended) loadings, although in many cases real loadings are lower than stated by the sample name.

*3.4. Characterization Methods*

Different methods were used to characterize the prepared catalysts. The absorption spectrum, chemical identity, real loading amounts, oxidation states, morphology as well as size and distribution on the supporting material was of interest. Below is a list of all used methods, a short description of each, and the reason why the method was applied.

### 3.4.1. Attenuated-Total Reflection Fourier-Transformed Infrared Spectroscopy (ATR-FTIR)

ATR-FTIR is a reliable spectroscopic method for fingerprint recognition of IR-absorbing materials. This method enables the recognition of characteristic functional groups in solid, liquid or gaseous samples [48]. To qualitatively confirm the phosphate-modification of $TiO_2$ and for the investigation of the Cu- and Ni-impregnated samples, an ATR-FTIR spectrometer (Bruker Tensor 27, Ettlingen, Germany) was used with a diamond crystal accessory. The samples were measured in the solid-state.

### 3.4.2. Diffuse Reflectance Spectroscopy (DRS)

DRS is referred to as UV-Vis spectroscopy in solid-state and can be used to evaluate the absorption properties of supported thin films and free-standing powders. In a single measurement, a thin film of the powdered sample was formed using a designated sample holder and its reflectance profile was recorded in the range between 200 and 800 nm relative to $MgSO_4$ reference using Jasco 670 spectrometer (Tokyo, Japan) and 2 mm sample holder. Absorbance (A) has been calculated from reflectance (R) values with: $A = 1/[\log (R/100)]$ and is plotted in arbitrary units (a.u.).

### 3.4.3. Total Reflection X-Ray Fluorescence Spectroscopy (TXRF)

TXRF is a powerful qualitative and quantitative technique for chemical multi-elemental analysis. Mainly liquid samples, but also solids, can be analyzed with detection limits in the ppb range [49]. Functionalization of $TiO_2$ with $PO_4$-groups, concentrations of co-catalysts in washing solutions and co-catalyst loading amounts can be determined through this method.

Quantitative chemical analysis to determine the metal loadings (wt.%) of solid nanocomposites was performed with Total Reflection X-ray Fluorescence (TXRF) using an ATOMIKA 8030C X-ray fluorescence analyzer (Atomika Instruments GmbH, Oberschleissheim, Munich, Germany). A Mo X-ray tube was used for sample excitation (monochromatized K$\alpha$-line) at 50 kV and 47 mA for 100 s, using the total reflection geometry and an energy-dispersive Si(Li)-detector. 1 mg of the samples were fixated at the center of quartz reflectors by pipetting 5 μL of a 1% polyvinyl alcohol (PVA) solution and drying for 5 min on a hot plate. Unloaded reflectors were measured beforehand to account for true blanks and after every set of measurements to rule out contaminations in the analyzer. Ti was set as the matrix with 100% and relative amounts of the loaded element of interest were acquired (wt.%).

For liquid samples, a Wobistrax TXRF analyzer was employed in a dual energy-band excitation experimental setup [50]. The source was a Rh X-ray tube (monochromatized K$\alpha$-line), at V = 50 kV and I = 0.7 mA. Measuring time was also 200 s, total reflection geometry and an Si detector were implemented. The liquid samples were added a Ga/Y internal standard (10 ppm) for quantification. They were pipetted (5 μL) on the quartz reflectors and dried on a hot plate.

### 3.4.4. Powder X-Ray Diffraction (XRD)

Powder X-ray diffraction (XRD) was performed using an XPERT II: PANalytical XPert Pro MPD (θ–θ Diffractometer, Panalytical, Almelo, The Netherlands). The sample was irradiated using a Cu X-ray source (8.04 keV, 1.5406 Å) after adding a standard reference Si-Powder (for line position and shape) and placing on a Si sample holder. Bragg–Brentano θ/θ-diffractometer geometry was employed, and the signals were acquired from 10 to 120 degrees. The detector system used was a semiconductor X'Celerator (2.1°) detector. The peak positions of the acquired XRD spectra were corrected with respect to the Si reference, and Rietvield refinement was performed to obtain the lattice parameters for all the phases.

### 3.4.5. Scanning Electron Microscopy (SEM)

Scanning electron microscopy (SEM) images were acquired using FEI Quanta 250 FEG (Hillsboro, OR, USA) at 200 keV scanning electron microscope to obtain information on the

morphology of the samples. Typically acceleration voltage of 5 kV and secondary electron detection mode were used.

### 3.4.6. Transmission Electron Microscopy (TEM)

TEM is an important technique for nanomaterial characterization. It allows the imaging and (chemical) characterization of materials on a nanometer scale. Hereby, an electron beam goes through a thin sample, causing elastic and inelastic scattering interactions which get magnified through electromagnetic lenses. Transmission electron microscopy (TEM) and high-resolution TEM (HRTEM) images were recorded on a Fei Tecnai F20 transmission electron microscope operating at 200 kV.

### 3.5. Photocatalytic Tests

To test the photocatalytic activity of the prepared composites, hydrogen evolution reaction (HER) experiments were conducted. In a single experiment, a sample of interest (10 mg) was sonicated in a 40 mL 1:1 mixture of water and methanol (used here as a sacrificial electron donor) for 60 s. The suspension was then transferred to the water-cooled reactor and it was purged with argon (100 mL min$^{-1}$) for 5 min to deaerate the reactor volume and remove dissolved gases ($O_2$, $CO_2$) under stirring at 300 rpm. Subsequently, the reactor was closed with a quartz glass window held by a metal lid. Argon was flushed for a further 5 min and then the flow was changed to 30 mL/min, at which point the data acquisition started. After approximately 30 min of flow stabilization, the UV-lamp (365 nm) was turned on. For most experiments, the illumination time was set to 30 min, after which the light was shut off, data acquisition stopped and the experiment was terminated.

The reactor was placed in a dark-box to avoid interaction with any other light source. Argon flows were regulated by a mass flow controller and the hydrogen evolution was detected in real-time using an Emerson X-Stream gas analyzer capable of detecting $H_2$-concentrations in the gas phase (in ppm) by a thermal conductivity detector. Besides $H_2$, the X-Stream analyzer (Rosemount Analytical Inc, Anaheim, CA, USA) is also capable of detecting $CO_2$, $Cl_2$ and $O_2$ with IR-, UV- and paramagnetic detectors, respectively.

Efficiency of a Photocatalyst

The HER-activity recorded by the X-Stream analyzer was given in ppm and was transformed with the aid of the ideal gas Equation (1) and the argon flow rate (in mL min$^{-1}$) to mols of $H_2$ that were produced per time unit. This activity will be further referred to as "$H_2$ evolution rate" and is given in μmol h$^{-1}$. Our following conversions used the resulting Equation (2). The value at the denominator arises from the standard conditions used: $T = 298$ K, $p = 101.33$ kPa and $R = 8.314$ m$^3$ Pa mol$^{-1}$ K$^{-1}$.

$$pV = nRT \tag{1}$$

$$H_2 \; evolution \; rate \; \left(\mu mol \; h^{-1}\right) = \frac{activity \; (ppm) \cdot flowrate \; \left(mL \; min^{-1}\right)}{407.5} \tag{2}$$

This $H_2$ evolution rate, while useful for describing the overall activity of the composite catalyst, does not account for the catalytically active sites. To be able to qualify the efficiency of a specific and catalytically active co-catalyst species, further parameters need to be taken into consideration. The turnover frequency (TOF), which made its first appearance in hetero- and homogeneous catalysis, first as a borrowed term from enzymatic kinetics, is still a term of debate, which is why a precise definition is very important [51]. We adopt a similar TOF definition to the one that Ye et al. have proposed: "the ratio of the number of the molecules produced per unit time at a single active site" [52]. The TOF of our catalysts is calculated as in Equation (3):

$$\mathrm{TOF} \left(h^{-1}\right) = \frac{number \; of \; produced \; H_2 \; molecules \; (mol)}{number \; of \; active \; co-catalyst \; sites \; (mol) \cdot time \; (h)} \tag{3}$$

TOF values are convenient to be estimated in homogeneous catalytic systems. Calculating the TOF value of a heterogeneous catalyst is rather difficult because the actual number of catalytically active sites is not usually defined accurately. However, since HSM-SCs are bridging homo- and heterogeneous catalysis, we define our "number of active co-catalyst sites" as the "total amount of co-catalyst atoms on the composite" (estimated through TXRF spectroscopy). For high loadings of co-catalyst species, which will likely form nanoparticles, the actual number of active sites will be less, since only the atoms at the surface of a nanoparticle would take part in the photocatalytic reaction, increasing the TOF-values. However, for the sake of comparability with the lower loadings systems, in which a better atom utilization is expected, the number of co-catalyst atoms will be considered as the number of active sites throughout this work.

## 4. Conclusions

Two series of Cu and Ni photocatalysts were prepared on bare and phosphate modified $TiO_2$ through a wet impregnation method following the concept of the site-isolation. Through various characterization techniques, we clarified the possible states and binding sites of the co-catalyst species, as well as the crystallinity, structure and morphology of the prepared photocatalysts. Photocatalytic HER data expressed in terms of TOF numbers were compared to those predicted by geometrical models. Reflected in the non-linear increase of the TOF values as a function of co-catalyst loadings (Model 2), the data for both earth-abundant systems suggested a tendency of the co-catalyst species to undergo a strong change in the shape and size, which indicates the success of the site-isolation strategy. Based on the increased TOF enhancement factors (14 to 39 in the case of Cu), the use of the $PO_4/TiO_2$ support was shown to further facilitate the morphological change towards smaller species, thus confirming the beneficial structural effect of surface modification on single-site stabilization. Overall, a maximal TOF values of up to 657 $h^{-1}$ and 1077 $h^{-1}$ were achieved for Ni and Cu on bare $TiO_2$. In the case of the $PO_4$-modified support, the maximal TOF for $Cu/PO_4/TiO_2$ even surpassed maximal TOF for the analogous reference Au-based sample. This study provides a proof-of-principle on how correlating real loadings to catalytic performances and comparing these trends to geometrical models can shed light on the structural rearrangement of the active co-catalyst species. It further illustrates the feasibility of the site-isolation strategy towards downsizing the co-catalyst species and highlights the importance of surface chemistry for their stabilization. We believe that these findings will apply to other noble- and non-noble-metal co-catalyst systems on the way to the exploration of the concept of single-site photocatalysis.

**Supplementary Materials:** The following are available online at https://www.mdpi.com/2073-4344/11/4/417/s1, Figure S1: Precursor species in liquid solutions after impregnation, Figure S2: Impact of precursor on ultimate loadings, Figure S3: ATR-FTIR spectra of the as-prepared photocatalysts, Figure S4: ATR-FTIR spectra of the used support materials, Figure S5: Absorbance spectra obtained through DRS measurements, Figure S6: SEM images for morphology comparison, Figure S7: TEM images for the investigation of foreign co-catalyst nanoparticles, Figure S8: TOF values of reference Au-samples against intended loadings.

**Author Contributions:** Investigation, Methodology, Visualization, Writing—original draft, P.A.; Investigation, A.G.; Investigation, S.P.N.; Investigation, S.N.M.; Investigation, P.W.; Conceptualization, Supervision, Writing—Review and Editing, Project Administration, Funding Acquisition, A.C. All authors have read and agreed to the published version of the manuscript.

**Funding:** A part of this research was funded by the Austrian Science Fund (FWF), project number P32801.

**Acknowledgments:** The authors would like to acknowledge the facilities of the TU Wien for technical support and fruitful discussions: Atominstitut (ATI) and especially Christina Streli and Peter Kregsamer; X-Ray Center (XRC) and especially Werner Artner; Analytical Instrumentation Center (AIC); and Electron Microscopy Center (USTEM). The authors would like to thank Jasmin Schubert

for the test XPS measurements. The authors acknowledge the TU Wien Bibliothek for financial support through its Open Access Funding Program as well as the Austrian Science Fund (FWF).

**Conflicts of Interest:** The authors declare no conflict of interest.

## Appendix A

*Model Calculations*

The mathematical reasoning behind Model 1 (Figure A1a) is the following: The $H_2$ evolution rate is directly proportional to the surface atoms of the co-catalyst nanoparticles (NPs) that are involved in the catalytic reaction. Decreasing the co-catalyst loading without changing the size/shape of the NPs results in a linear decrease of the number of particles, i.e., a linear decrease of the surface atoms and therefore a linear dependency ($y = x$) of the loading vs. the HER.

$$If: \quad H_2 \; evolution \; rate \; \sim \; surface \tag{A1}$$

$$and \quad surface \; \sim \; co\text{--}catalyst \; loading \tag{A2}$$

$$Then: \; H_2 \; evolution \; rate \; \sim \; co\text{--}catalyst \; loading \tag{A3}$$

The turnover frequency, defined as

$$TOF = \frac{H_2 \; evolution \; rate}{co\text{--}catalyst \; moles} \tag{A4}$$

will be a constant, since:

$$H_2 \; evolution \; rate \; \sim \; co\text{--}catalyst \; loading \tag{A5}$$

Subsequently, the HER will be divided by the loading and hence:

$$TOF = constant. \tag{A6}$$

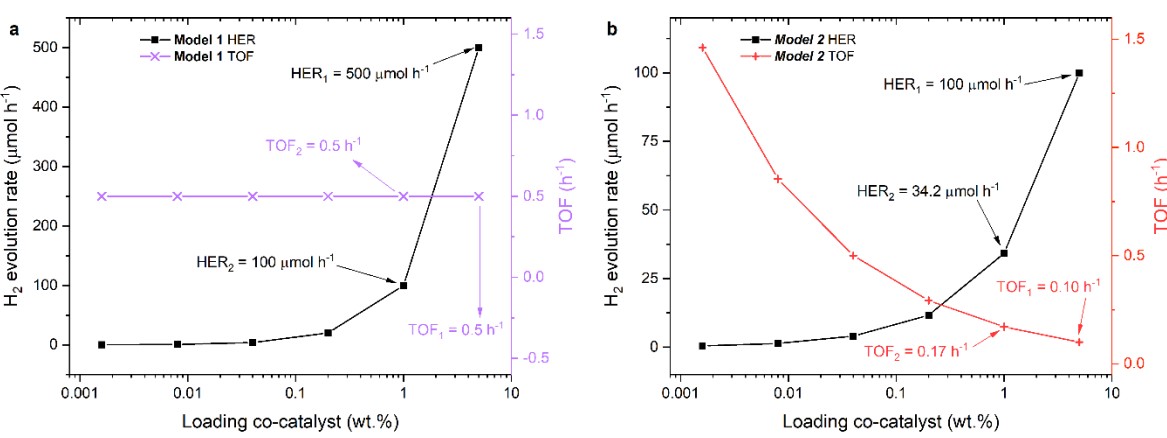

**Figure A1.** Graphical representations for (**a**) Model 1: loading vs. HER (black slope, $y = x$) and loadings vs. TOF (purple constant, $y = a$ ) and (**b**) Model 2: loading vs. HER (black line, $y = x^{2/3}$ ) and loading vs. TOF (red line, $y = x^{-1/3}$ )

As an example, arbitrary initial values will be taken for showing the behavior of Model 1 graphically (Figure A1a): a loading of 5 wt.% (supposing it is 1000 µmol atoms at the surface of the particle that are catalytically involved in the reaction) would give an activity of 500 µmol $h^{-1}$.

Decreasing the loading by a factor of 5 would result in a decreased number of surface atoms involved in the HER (black) by the same factor. Since the size/morphology of the co-catalyst particle stays unchanged, but its number gets reduced (linear decrease of surface atoms with the loading). This means for 1 wt.% (200 µmol) the $H_2$ evolution rate

will also decrease by a factor of 5, i.e., 100 μmol h$^{-1}$. The TOF (purple), in this case, is a constant, since:

$$\text{TOF} = \frac{\text{H}_2 \; evolution \; rate}{co-catalyst \; moles} = \frac{500 \; \mtext{μmol} \; h^{-1}}{1000 \; \mu mol} = \frac{100 \; \mu mol \; h^{-1}}{200 \; \mu mol} = 0.5 \; h^{-1} \qquad (A7)$$

The same reasoning applies when going towards lower loadings.

In the case of Model 2 (Figure A1b), we assume that the size/morphology of the co-catalyst NPs changes. The surface (*S*) and volume (*V*) of a sphere with a radius (*r*) are:

$$S = 4 \; \pi \; r^2 \;\; and \;\; V = \frac{4}{3} \; \pi \; r^3 \qquad (A8)$$

Setting *S* as a function of *V* (i.e., loading) and substituting *r* results in the following equation:

$$S_{(V)} = 4\pi \left( \frac{3V}{4\pi} \right)^{\frac{2}{3}} \qquad (A9)$$

Upon simplifying we obtain:

$$S_{(V)} = \left( 6\sqrt{\pi} \; V \right)^{\frac{2}{3}} \qquad \rightarrow \qquad S_{(V)} \sim V^{\frac{2}{3}} \qquad (A10)$$

Since

$$HER \sim S \quad and \quad loading \sim V \qquad (A11)$$

then, the same relation as for *S* and *V* follows for *HER* and *Loading*:

$$\text{H}_2 \; evolution \; rate \sim loading^{\frac{2}{3}} \qquad (A12)$$

The TOF, as a function of the loading follows a different proportionality due to its definition:

$$\text{TOF} = \frac{\text{H}_2 \; evolution \; rate}{co-catalyst \; moles} \sim \frac{loading^{\frac{2}{3}}}{loading} \sim \frac{1}{loading^{\frac{1}{3}}} \;\; \text{TOF} \sim loading^{-1/3} \qquad (A13)$$

For example: assuming a sample with 5 wt.% (1000 μmol atoms) and an initial HER$_1$ value of 100 μmol h$^{-1}$. A decrease by a factor 5 in the loading would then have a decrease in the HER by a factor of $(x)^{2/3}$, i.e., we expect that 1 wt.% (200 μmol atoms) would render a HER$_2$ of

$$\text{HER}_2 = 100 \; \mu mol \, h^{-1} \div (5)^{2/3} = 34.2 \; \mu mol \; h^{-1} \qquad (A14)$$

The TOF, in turn, would follow the proportionality of $(x)^{-1/3}$. Differently to Model 1, where the TOF was a constant, in the case of Model 2 we observe an increase of the TOF (red curve) with decreasing loadings:

$$\text{TOF}_1 = \frac{100 \; \mu mol \; h^{-1}}{1000 \; \mu mol} = 0.10 \; h^{-1} \neq \text{TOF}_2 = \frac{34.2 \; \mu mol \; h^{-1}}{200 \; \mu mol} = 0.17 \; h^{-1} \qquad (A15)$$

Figure A2 is obtained by plotting the expected TOF trends in a double logarithmic representation, which can be used to qualitatively compare the behavior of the investigated series of prepared composite photocatalysts.

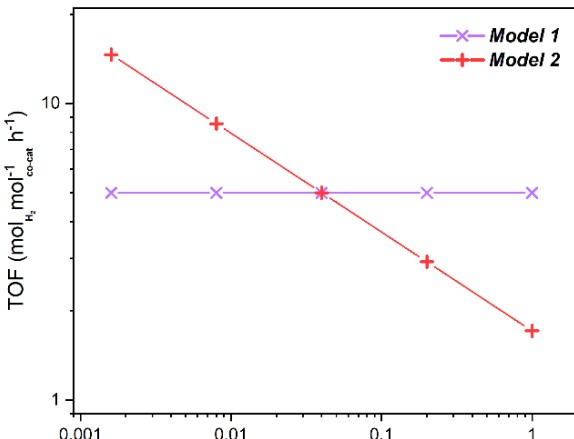

**Figure A2.** Double logarithmic representation of the TOF trends for Model 1 (purple constant) and Model 2 (red slope) for comparison with the catalytic performance of the series-prepared composite photocatalysts.

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
