# Peer review of "Isolation Strategy towards Earth-Abundant Single-Site Co-Catalysts for Photocatalytic Hydrogen Evolution Reaction"

_catalysts, doi:10.3390/catal11040417_

Round 1

Reviewer 1 Report

The article entitled "Isolation strategy towards earth-abundant single-site co-catalysts for photocatalytic hydrogen evolution reaction" introduced the concept of heterogeneous single-metal-site catalysts (HSMSCs) and the idea of site-isolation. They synthesized various amounts of metal loading as co-catalyst and proposed an interesting structural model to explain the observed TOF trends. However, there are some parts needed to be clarified, as shown below:

1. The photocatalytic performance of Cu is much better than Ni. Why? The authors should explain. 
2. Please show the DRS spectra of Cu/PO4/TiO2 and Ni/PO4/TiO2 for comparison.
3. Page 5, line 160, the full name of SPR should be provided.
4. The shoulder in DRS suggested the Ni metal doping, which is not corresponding with the topic (metal loading). More evidence needs to be provided. 
5. The dispersion is very important in this research. The authors must inspect the Cu or Ni dispersion.   
6. It is weird to observe Ni nanoparticles only in the case of 5/Ni/TiO2 in TEM. The introduction of phosphate groups increases the metal loading significantly, so it should be capable of being observed.
7. How to calculate excess coverage? Authors must introduce or refer to some literature.
8. The brand of chemicals employed in this paper must be noted. 
9. The surface area of pure TiO2 (80m2/g) seems too high. 
10. Ni and Cu should be reduced to form metallic properties. However, the reduction seems to be not mentioned in the synthesis procedure. 
11. XPS analysis is suggested to inspect the oxidation state of Ni and Cu.
12. The TXRF data should be shown, and it seems not so representative to be noted as the accurate metal loading, even TOF.  

Reviewer 2 Report

The authors presented in this study a strategy to fabricate heterogeneous single metal catalysts containing earth-abundant metals, such as Ni and Cu. The authors focused on pathways to specifically tailor the surface of the catalyst and control the loading and distribution of Ni or Cu on TiO2 nanoparticles. The study is well written and offers an interesting perspective on the synthesis of efficient catalysts in heterogeneous catalysis. I invite the authors to review part of their characterization strategy to further improve the quality of the current manuscript.

A minor revision is recommended on the basis of the following comments:

  • In terms of FTIR-ATR analysis, I wonder whether the authors have removed the baseline from their spectra. The bands shown in Figure 2 seems to be very weak and the analytical method used to analyze the spectra should be better clarified. Although I do believe that the bands shown are significant, there should be a stronger discussion revolving around the assessment of these spectra.
  • Please change the color scale in Figure 3, it is quite hard to differentiate between the different curves (see those in cyan/turquoise for instance)
  • I am not surprised to see that both XPS and XRD failed to provide more insight into the presence and/or oxidation state of the metals given the low content. I believe the authors should consider having EDS/TEM analysis of their catalysts to better show the distribution of Cu and Ni.
  • Have the authors considered XAS as a type of spectroscopy functional to identify the oxidation state of Cu and Ni?
  • Can the authors elaborate further on the performance of these catalysts? I understand this is not the main purpose of the study, but there is little to no benchmarking. How do these catalysts compare to the SACs synthetized from noble metals?

Reviewer 3 Report

Тhe paper deals with preparation and characterization of Cu - and Ni - composites for photocatalysis following the concept of single-site isolation. The materials here are carefully characterized using advanced physical methods: STP-IR, DRS, TEM, SEM and their efficiency has been demonstrated. The paper is interesting and can be published in CATALYSTS after MINOR REVISION. Questions /Q/ and remarks /R/ are listed bellow:

Q1 - what is the definition of "active catalitic sites", eq 3?

R1 Figure 3 is confusing: the output of DRS spectra is F(R) or R, not absorbance. Absorbance is = lg(I0/I) and can not be expressed in %. Please, redraw figure 3, and use F(R)!

Round 2

Reviewer 1 Report

The authors have carefully revised their manuscript according to the comments. I am satisfied with the responses and revisions.